# ParaRater: Enhancing Cross-Lingual Transfer in LLMs with Meta-Learning

## Abstract

Multilingual LLMs are rapidly emerging, accompanied by claims of supporting an ever-increasing number of languages. However, significant gaps remain between their performance in English and in other languages. Due to the limited quantity and quality of low-resource language data, independently improving these languages is a tough route. A natural alternative is to transfer the capabilities learned in English to low-resource languages. Parallel corpora play a key role in such transfer, and some prior works have conducted empirical studies. Yet, which types of parallel corpora contribute most effectively to cross-lingual transfer has not been systematically explored. To address this, we propose ParaRater, a corpus selection method designed to identify the most valuable English data to be translated into target languages, thereby constructing high-quality parallel corpora that efficiently boost performance in those languages. ParaRater leverages meta-learning to directly align corpus selection with model performance on native target-language data. It further employs a two-stage filtering process to pinpoint data that is only effective when both language versions appear in training—i.e., truly impactful parallel corpora. We demonstrate the effectiveness of this approach across multiple languages and provide detailed qualitative analyses, offering new insights into cross-lingual transfer in large language models. Our rater, datasets, and code are all released open-source. Our code and data are anonymously open at `https://anonymous.4open.science/r/dVqzLmtrwFae-8676`.

## 1 Introduction

Multilingual large language models (LLMs) are rapidly proliferating, touting support for an expanding array of languages. Gemma 3 (Team et al., 2025) offers support for over 140 languages, and Qwen 3 (Yang et al., 2025) expands its predecessor's capabilities by supporting 119 languages and dialects. On closed-source models, GPT-5 [1] and Claude-4 [2] also demonstrate broad multilingual coverage. Despite these advances, significant performance gaps persist between English and other languages—especially low-resource ones (Han et al., 2025). Methods for improving LLMs' performance in non-English languages mainly fall into two directions: one is to independently enhance target-language ability by increasing the quantity and quality of its training data (Zhao et al., 2025; Dou et al., 2025), and the other is to leverage the abundance of high-quality English data to transfer English capabilities to other languages (Shen et al., 2025; Fujii et al., 2024; Lu et al., 2023; Qorib et al., 2025).

What remains unexplored is a fundamental research problem: which types of parallel corpora truly drive cross-lingual transfer in LLMs, and how can we systematically identify them? Using parallel corpora to train models for cross-lingual transfer is a mainstream approach. Existing approaches typically treat all parallel data as equally useful, relying on heuristics or post-training adaptation, with benefits observed mostly in translation tasks. However, existing work has been largely empirical, focusing mainly on the post-training stage, with the primary benefits of parallel data observed in translation tasks. There has been little systematic investigation into how to select parallel corpora or how they influence broader general-purpose language capabilities. Furthermore, not all parallel corpora contribute in the same way—some English texts improve target-language performance even

---

[1] `https://openai.com/index/introducing-gpt-5/`
[2] `https://www.anthropic.com/news/claude-4`

when used alone, while others only exert impact when paired with their translations. Disentangling this distinction is crucial: only the latter represents the "core" of parallel corpora that genuinely induces transfer. This motivates us to design a principled framework to automatically identify and exploit the most impactful subsets for efficient multilingual pretraining.

To address these challenges, we propose ParaRater, which employs a meta-learning framework to directly align the impact of parallel corpora with LLMs' performance on a validation set. The goal of ParaRater is to identify the subset of parallel data that truly drives cross-lingual transfer—what we term the core of the parallel corpus. We define the core of the parallel corpus as the portion of source-language data that can only provide positive impact on the target language when paired with its translated counterpart. For example, in the case of English-Chinese, we aim to identify the most valuable English documents that can enhance an LLM's Chinese ability. These documents have little effect on improving Chinese performance when included alone in training, but once translated into Chinese and added as parallel data, their original English versions begin to contribute positively to Chinese capability.

To achieve this goal, we design a two-stage strategy to filter the core of the parallel corpus. Each stage employs a rater that assigns importance weights to documents with respect to their contribution in the target language. In the first stage, the rater selects documents that receive high weights in both the source and target languages. In the second stage, the target-language versions of these documents are removed from the training data, and the rater then identifies the subset of source-language documents whose weights drop significantly—these constitute the core documents. These documents represent the portion of parallel data that truly exerts its effect in a parallel manner, as opposed to pseudo-parallel data that exists in parallel form but whose benefits stem solely from the monolingual side. With this approach, we can make more efficient use of parallel data—for example, by extracting the core subset from existing parallel corpora, or by selecting high-value subsets of English data to be translated for constructing new parallel corpora. To control for confounding factors, we conduct experiments by pretraining 1.2B-parameter LLMs from scratch on multiple bilingual pairs. The results show that ParaRater significantly outperforms strong baselines. We further provide detailed analyses and case studies of the selected data, offering new perspectives on cross-lingual transfer in LLMs. To summarize, our contributions are as follows:

- We propose ParaRater, which leverages meta-learning and a two-stage strategy to disentangle the effective components of parallel data from pseudo-parallel data, providing a more efficient way to exploit parallel corpora.

- We construct the ParaCore training set, consisting of core parallel data between English and several other languages, which significantly improves LLM performance in non-English languages.

- We develop ParaTool, a meta-learning framework for training parallel-data selectors, which enables convenient identification of the most valuable data for cross-lingual transfer across arbitrary languages and datasets.

- We conduct controlled experiments on multiple bilingual pairs using clean parallel data, and provide both qualitative and quantitative analyses that reveal the role of parallel corpora in LLM pretraining, offering valuable insights into cross-lingual transfer.

## 2 RELATED WORK

**Cross-lingual Capability Transferring with Parallel Corpus**  Because the amount of high-quality data in non-English languages is significantly smaller than in English, and increasing the proportion of low-resource language data often leads to a decline in English performance, many studies have resorted to using parallel corpora to enhance LLM capabilities in low-resource languages. Although some studies have explored the impact of parallel corpora, the field is still far from being well-documented. Shen et al. (2025) introduce a method to scale multilingual LLMs by leveraging multi-way parallel corpora, aligning previously unaligned models through systematic parallel data utilization. Fujii et al. (2024) explore continual pre-training for cross-lingual adaptation, showing that targeted additional training can substantially enhance Japanese capabilities in multilingual LLMs. Lu et al. (2023) propose TRIP, a triangular document-level pretraining strategy that accelerates multilingual pretraining by exploiting parallel data triplets at the document level.

Qorib et al. (2025) demonstrate that systematically incorporating parallel corpora into training improves the multilingual capabilities of LLMs across diverse languages. These studies confine their exploration of parallel data either to the post-training stage or to its impact on translation capabilities. Yet the pretraining stage is the most critical phase for shaping multilingual competence in LLMs, and to the best of our knowledge, no prior work has systematically examined the influence of parallel corpora on LLMs' general capabilities in pretraining phase.

**Data Curating**    Selecting high-quality subsets from raw corpora for pretraining is a crucial step in determining LLM performance. The main approaches include heuristic rule-based filtering and LLM-as-judge methods. Heuristic filtering removes duplicates, overly short, or toxic texts using regular-expression matching (Laurençon et al., 2023; Weber et al., 2024; Penedo et al., 2023; Soldaini et al., 2024), or applies simple classifiers based on features such as perplexity or text type for selection(Chowdhery et al., 2022; Touvron et al., 2023a; Xie et al., 2023; Muennighoff et al., 2023). However, such methods are limited in generalizability and accuracy, and in recent years, LLM-as-judge approaches for text classification or quality assessment have become the mainstream. QuRating (Wettig et al., 2024) trains a model to evaluate texts along dimensions like writing style, expertise, facts/trivia, and educational value. FineWeb-Edu (Penedo et al., 2024) filters data with an LLM-based educational quality classifier trained on synthetic annotations from Llama-3. FIRE (Xu et al., 2025) aligns diverse data quality raters and introduce a progressive data selection scheme. MuRating (Chen et al., 2025) combines multiple English raters via pairwise comparisons to train a unified rater. Although these methods can filter corpora along more sophisticated dimensions than heuristic rules, the signals used to train the raters are still grounded in human-predefined criteria. DataRater (Calian et al., 2025) trains a rater model to score training samples directly aligning with the effectiveness in LLM pretraining, using meta-learning method. Inspired by this, we adopt a meta-learning approach to investigate the effectiveness of parallel corpora in LLM pretraining, and train raters to perform data selection.

## 3    PRELIMINARY: DATA SELECTION WITH META-LEARNING

Using meta-learning for data selection primarily involves training a rater that assigns scores to data, with the selected subset then used for model pretraining; the optimization objective is to minimize the loss of the pretrained model on a given test dataset.

Given a training corpus $\mathcal{D}_{\text{train}}$, a rater model $R_\eta$ is used to score the documents $d \in \mathcal{D}_{\text{train}}$. A language model $M_\theta$ is trained with the curated data and a loss function $l$. The gradient of the language model in a step $t$ is defined as:

$$g_t = \frac{1}{n} \sum_{i=1}^{n} \nabla_\theta l(d_i; \theta_t), \tag{1}$$

where $d_i \in R_\eta(\mathcal{D}_{\text{train}})$ is a batch of documents selected by the rater at step $t$. After $T$ steps, the optimized parameters are obtained, which is denoted as

$$\theta^* = \theta(R_\eta(\mathcal{D}_{\text{train}})).$$

The optimized language model is then evaluated by a loss on a given test dataset $L(\theta_t; \mathcal{D}_{\text{test}})$. The optimization objective of the rater $R$ is:

$$\eta^* = \arg\min_\eta L(\theta(R_\eta(\mathcal{D}_{\text{train}})); \mathcal{D}_{\text{test}}) \tag{2}$$

The task is to filter a predefined size of subset $\mathcal{D}'$, where the documents in it are with high effectiveness in pretraining the language model.

## 4    PARARATER

Unlike typical data selection tasks, the goal of ParaRater is to identify the high-value subset from source data to construct target-language corpora, thereby forming parallel data. The key difference

is that we not only aim to identify the portion of the corpus that contributes to improving the target-language capability, but also to disentangle the bilingual dependency within that portion.

In Section 4.1, we first formulate the problem of parallel data selection and present the overall optimization objective. Since directly optimizing this objective is computationally infeasible, in Section 4.2 we approximate it using a two-stage procedure. In the first stage, we identify documents in the parallel corpus that provide benefits in both languages. In the second stage, we keep only the source-language documents and remove their target-language counterparts, then compare the scoring differences between the two raters across the two stages to identify the core of parallel data—the documents whose bilingual coupling is the strongest.

## 4.1 PARALLEL CORPUS FILTERING PROBLEM

Given a training dataset in the source language $\mathcal{D}_s$ and a test dataset $\mathcal{D}_{\text{test}}$ in the target language, the rater selects a high-value subset of pre-determined size

$$\mathbf{d}_s = R(\mathcal{D}_s).$$

After the selection, $\mathbf{d}_s$ is used to build its target language version $\mathbf{d}_t$, forming the parallel corpus. A language model is then optimized on the developed parallel corpus. We rewrite Eq. 1 as

$$g = \nabla_\theta l(\mathbf{d}_s \cup \mathbf{d}_t; \theta), \tag{3}$$

where we omit the step annotation for simplicity. Following the approach in section 3, we use the Eq. 2 to optimize the rater, with the $R(\mathcal{D}_s)$ replaced by $\mathbf{d}_s \cup \mathbf{d}_t$.

However, this objective does not guarantee that the constructed subset achieves the highest efficiency for language capability transfer. Parallel corpus that can drive effective cross-lingual transfer should manifest in the way that the two language versions of the same data mutually influence their effectiveness during model training. What we seek is to identify cases where the combination of two languages produces high value when they appear together, rather than data that is already highly valuable in a monolingual setting. Therefore, we need to introduce additional constraints. The final optimization objective for the rater is then as follows:

$$\eta^* = \arg\min_\eta \Big( L(\theta(\mathbf{d}_s \cup \mathbf{d}_t)) - L(\theta(\mathbf{d}_s \cup \mathbf{d}'_t)) \Big), \tag{4}$$

where $\mathbf{d}'_t$ denotes the target-language counterpart that is mismatched with $\mathbf{d}_s$ and $\mathbf{d}'_t$ satisfies $L(\theta(\mathbf{d}_t)) = L(\theta(\mathbf{d}'_t))$, i.e., $\mathbf{d}'_t$ is of equal quality to $\mathbf{d}_t$. We omit the explicit dependence of $L$ on $\mathcal{D}_{\text{test}}$. In this objective, the selected subset of source corpus only contributes to the language model performance when the model is jointly trained on the subset and its target-language counterpart.

## 4.2 DUAL-RATER CROSS FILTERING

Since the selection from the corpus is a discrete sampling process, directly optimizing Eq. 4 is a non-deterministic polynomial-time hard problem. Inspired by DataRater (Calian et al., 2025), we instead optimize the rater by assigning continuous weight values to batches of training samples. Moreover, directly optimizing Eq. 4 would require maintaining multiple language models and computing their gradients simultaneously, which is computationally sophisticated. Therefore, we adopt a two-stage strategy, optimizing two raters to cross-filter the data as an approximation. Figure 1 depicts the two stages.

**Stage 1: High-Value Bilingual Subset Selection** At this stage, we train rater $R_{\eta_1}$ to optimize the first term of Eq. 4, in order to select the documents that get high rank in language model pretraining in both languages. Here we have

$$\eta_1 = \arg\min_\eta \Big( L(\theta(\tilde{\mathbf{d}}_s \cup \tilde{\mathbf{d}}_t)), \Big)$$

where $\tilde{\mathbf{d}}_s$ and $\tilde{\mathbf{d}}_t$ represent a coarse-grained filtering of $\mathbf{d}_s$ and $\mathbf{d}_t$. In practice, we first construct parallel data $\mathcal{D}_t$ in the target language from the source language data $\mathcal{D}_s$. Then, the language model is trained on this parallel corpus and the rater meta-learns from the loss $L$ of the language model parameters on the test set $\mathcal{D}_{\text{test}}$ as the optimization objective. We then use the trained rater to score

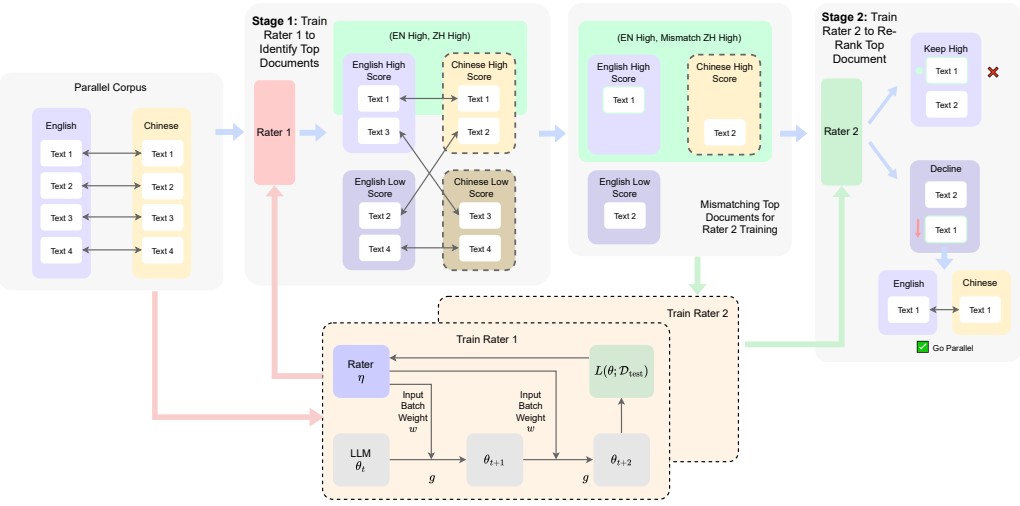

Figure 1: ParaRater training process. In Stage 1, Rater 1 is trained to select the top-performing documents in both languages. Using the scores from Rater 1, we then reorganize the parallel corpus by deliberately mismatching the top documents from the two languages. In Stage 2, Rater 2 is trained on this mismatched data to identify the source-language documents whose rankings drop once their target-language counterparts are removed. These documents—and their corresponding target-language versions—constitute the most valuable parallel data for pretraining.

$\mathcal{D}_t$ and $\mathcal{D}_s$ independently, and select the samples that rank high in both languages. In summary, at this stage, we performed a standard meta-learning procedure to filter out high-quality data from a training set that is composed entirely of parallel corpora, based on the model's performance in the target language.

**Stage 2: Low-Value Monolingual Source Language Subset Selection**    In the previous stage, although we successfully identified bilingual data that are highly valuable for improving performance in the target language, this does not necessarily imply effective language capability transfer—the language model may develop its capabilities independently in each language. The essence of parallel data lies in enabling interoperability: when the model learns knowledge in one language, that knowledge should interact with its understanding of the same content in the other language. For example, if learning a particular Chinese document improves the model's performance in Chinese, then learning the English version of that document should likewise bring improvements. In the context of rater meta-learning, this is reflected by the rater's score for the English document being influenced by the presence of its corresponding Chinese counterpart.

At this stage, our goal is to identify source-language data that improves model performance only when its target-language counterpart is also present. To do this, we start with the source-language documents that ranked highly in both languages in the previous stage. We then construct a new training set by combining:

- these high-ranking source documents,
- high-ranking target-language documents that are not parallel to them, and
- a small subset of low-ranking source documents (as a comparison group).

Using this mixture, we meta-learn a second rater and examine how the rankings of the high-scoring source documents change. If a document's ranking drops substantially, it indicates that the document is only useful when paired with its target-language counterpart. These documents—together with their corresponding target-language versions—form the core of the parallel corpus. Their performance drop in the absence of their paired translations reveals the strong cross-lingual dependency encoded in these parallel pairs.

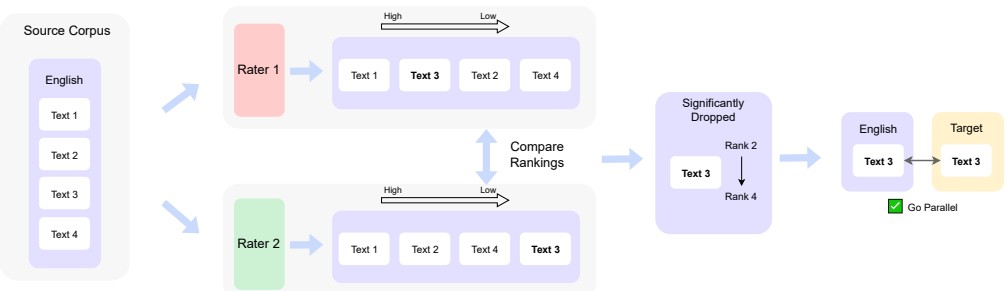

Figure 2: Source-only selection. Score the source-language data using Rater 1 and Rater 2, compare the differences in their scores, and identify the documents with significant drops to construct the core of parallel data.

Formally, a new rater $R_{\eta_2}$ is meta-trained on the high-value source language subset $\tilde{\mathbf{d}}_s$, the high-value mismatched target language subset $\tilde{\mathbf{d}}'_t$, and the low-value source language subset $\tilde{\mathbf{d}}'_s$. Following Eq. 3 the language model optimization is denoted as

$$g = \nabla_\theta l(\tilde{\mathbf{d}}_s^* \cup \tilde{\mathbf{d}}_t'^* \cup \tilde{\mathbf{d}}_s'^*; \theta), \tag{5}$$

Where $\tilde{\mathbf{d}}_s^* = R_{\eta_2}(\tilde{\mathbf{d}}_s)$, $\tilde{\mathbf{d}}_t'^* = R_{\eta_2}(\tilde{\mathbf{d}}'_s)$, and $\tilde{\mathbf{d}}_t'^* = R_{\eta_2}(\tilde{\mathbf{d}}'_t)$ are the subsets selected by the rater 2. Note again that this is an iterative process, and for simplicity we have omitted the step indices. The rater optimization is then defined as:

$$\eta_2 = \arg\min_\eta \left( L(\theta(\tilde{\mathbf{d}}_s^* \cup \tilde{\mathbf{d}}_t'^* \cup \tilde{\mathbf{d}}_s'^*)) \right).$$

$\tilde{\mathbf{d}}_s^*$ represents the source-language samples that remain highly valuable despite lacking a corresponding target-language counterpart. Since these samples already enhance the language model's capability in the target language on their own, there is no need to use them as seeds for constructing additional parallel data. Our focus is on

$$\mathbf{d}_s = \tilde{\mathbf{d}}_s - \tilde{\mathbf{d}}_s^*,$$

which is the core of the parallel corpus. This subset of source-language data whose usefulness relies on the presence of their parallel target-language counterparts.

**Source-only Selection** Once the two raters described above have been obtained, we can filter the existing parallel corpus to extract its core portion. However, naturally occurring parallel data are relatively scarce, and performing large-scale translation to construct large candidate parallel corpora is also computationally expensive. Therefore, a more important application is to first select the high-value subset from the source-language corpus and then translate it to form core parallel data. This can be easily achieved using ParaRater. Figure 2 depicts the source-only selection process.

In source-side filtering, the only difference from the two stages described above is that we do not have access to the rankings on the target-language side. As a result, stage 1 may include samples that rank highly in the source language but poorly in the target language. Such cases are primarily caused by low-quality or erroneous translations. The impact of these samples can be mitigated by improving translation quality.

## 5 EXPERIMENT

In this section, we conduct experiments under the Source-only Selection setting to evaluate the effectiveness of ParaRater in improving multilingual capabilities.

### 5.1 SETUP

**Dataset** Since naturally occurring parallel data are relatively scarce, we use machine-translated parallel data when training the rater. Unlike some previous work, we do not use concatenated par-

allel pairs in order to eliminate potential bias introduced by the data format; instead, all parallel documents are distributed independently within the training data. The experiment is conducted on multilingual data drawn from **Fineweb-2** (Penedo et al., 2025). For rater training, we sample 20B tokens from English corpora and translate them into 8 languages — Chinese, German, French, Japanese, Thai, Arabic, Tagalog and Russian — thereby ensuring coverage across multiple linguistic typology dimensions. Translations are generated using the Qwen3-8B [3] model, and documents with COMET (Rei et al., 2020) scores above 0.8 are retained to control and maintain high translation quality. We sample 2B tokens from the native corpus of each of the nine target languages to serve as the validation set in the meta-learning process.

To evaluate the effectiveness of ParaRater, we use the trained ParaRater for each language to perform source-side filtering on the FineWeb English data, selecting 1B tokens for each language. These selected samples are then translated into the corresponding target languages to form the core parallel corpus. We then add an additional 100B tokens of English data and pre-train a 1.2B-parameter language model based on the LLaMA architecture. This unbalanced design is intended to simulate the scenario where target-language data are extremely scarce, allowing us to evaluate the effect of transferring knowledge from English through translated parallel data.

**Baselines**   We compare ParaRater to other data selection methods, including **FineWeb-Edu Classifier** (Penedo et al., 2024), which is trained on LLM labeled data to identify educational content, and **DCLM** (Li et al., 2024), a FastText classifier trained to distinguish high-quality data from raw web data. We apply these data selection methods to select 1B tokens from the English corpus, translate them into the target languages, and pre-train the language models using the resulting parallel data combined with the same 100B tokens of English data as above.

**Evaluation**   We evaluate the pre-trained language models on both English and the target languages. For English evaluation, we adopt **ARC-E** (Clark et al., 2018), **Hellaswag** (Zellers et al., 2019), **StoryCloze** (Mostafazadeh et al., 2016), **BMLAMA** (Qi et al., 2023), **MMLU** (Hendrycks et al., 2021), covering multiple capabilities ranging from natural language understanding, commonsense reasoning, and factoid knowledge. For evaluation on the target languages, the widely used multilingual version of **Hellaswag** (Lai et al., 2023), **StoryCloze** (Lin et al., 2022) and **MMMLU** [4] are adopted. For **BMLAMA**, we observed that the original multilingual version contained a substantial number of errors, so we used the MuBench (Han et al., 2025) version instead. For the languages not covered by the above multilingual version, we also supplement them using MuBench. In addition, we use **FLORES** (Goyal et al., 2022) ChrF++ scores to evaluate the EN-to-target translation capabilities of the models.

**Implementation**   We conducted both rater training and language model training on a cluster of 64 H100 GPUs. To avoid interference from cross-lingual factors and ensure fairness, we train a separate ParaRater for each En–Target language pair. For each language, ParaRater training consumed roughly 24 × 64 H100 GPU-hours, while training each final language model required 12 × 64 GPU-hours. As the initial rater, we used Qwen3-Embedding-0.6B [5] with a linear head. The language model co-trained with the rater during meta-learning, as well as the final language model, share the same architecture with 1.2B parameters. Inspired by DataRater, we developed **ParaTool**, a meta-learning toolkit for rater training, and applied several optimizations to make large-scale training practical. Both **ParaTool** and **ParaCore**—the parallel corpus we constructed through ParaRater—are open-sourced to foster further research on multilingual large language models.

## 5.2 Results

Table 1 shows the result. We observe that ParaRater demonstrates clear advantages on most benchmarks, covering both NLU tasks (e.g., HellaSwag, StoryCloze, xWinograd) and factual knowledge tasks. Moreover, ParaRater shows substantial improvements on translation tasks, aligning with prior observations that parallel corpora significantly boost translation performance. Compared with other

---

[3] https://huggingface.co/Qwen/Qwen3-8B

[4] https://huggingface.co/datasets/openai/MMMLU

[5] https://huggingface.co/Qwen/Qwen3-Embedding-0.6B

data selection approaches for constructing parallel corpora, ParaRater achieves higher data efficiency, delivering notable performance gains using only 1B tokens per language.

We also note that ParaRater underperforms on science-oriented tasks (e.g., ARC-E). This may be related to the choice of validation set: we directly use multilingual data from FineWeb-2 as validation data, which results in a more comprehensive optimization target for ParaRater. In contrast, FineWeb-Edu explicitly targets the selection of educational content, which makes it more advantageous on science-focused benchmarks. Nevertheless, ParaRater still delivers a measurable improvement on English, suggesting that its data selection process effectively filters out noisy data introduced by low-quality translations, thereby preventing negative impacts on the model's English capabilities.

Table 1: Model performance trained on parallel data selected by different methods.

| | DCLM | | | | | FineWeb-Edu | | | | | ParaRater | | | | |
|---|---|---|---|---|---|---|---|---|---|---|---|---|---|---|---|
| **ARC-E** | EN 58.41 | ZH 48.28 | JA 45.74 | FR 44.89 | TH 39.85 | EN 59.52 | ZH **54.85** | JA **49.43** | FR **48.28** | TH **45.36** | EN **62.23** | ZH 49.81 | JA 45.82 | FR 45.82 | TH 41.42 |
| | TL 41.08 | DE 44.47 | RU 42.43 | AR 38.83 | | TL **42.31** | DE **47.99** | RU **45.70** | AR **42.94** | | TL 39.93 | DE 46.20 | RU 43.92 | AR 38.58 | |
| **MMMLU** | EN 28.93 | ZH 27.34 | JA 27.10 | FR 27.40 | TH 26.59 | EN 29.66 | ZH **27.88** | JA **27.74** | FR **27.98** | TH **27.28** | EN **29.80** | ZH 27.63 | JA 26.97 | FR 27.41 | TH 26.78 |
| | TL 27.58 | DE 27.21 | RU 26.55 | AR 25.89 | | TL 28.11 | DE 27.33 | RU 26.65 | AR 26.12 | | TL **28.38** | DE 27.33 | RU **26.90** | AR **26.33** | |
| **BMLAMA** | EN 69.35 | ZH 43.42 | JA 36.15 | FR 55.22 | TH 32.80 | EN 69.28 | ZH **46.43** | JA 40.39 | FR 57.23 | TH 34.96 | EN **70.04** | ZH 44.95 | JA **41.06** | FR **57.85** | TH **39.30** |
| | TL 63.88 | DE 60.11 | RU 35.24 | AR 38.55 | | TL **64.03** | DE **63.07** | RU 39.48 | AR 41.29 | | TL 63.91 | DE 62.81 | RU **39.78** | AR **43.22** | |
| **HellaSwag** | EN 53.51 | ZH 36.28 | JA 32.09 | FR 38.13 | TH 32.30 | EN **55.15** | ZH 38.32 | JA 32.80 | FR 40.14 | TH 33.71 | EN 55.13 | ZH **40.85** | JA **35.65** | FR **42.29** | TH **34.11** |
| | TL 36.47 | DE 34.14 | RU 36.17 | AR 33.18 | | TL 38.62 | DE 36.26 | RU 37.85 | AR 35.50 | | TL **39.22** | DE **37.35** | RU **40.37** | AR **37.01** | |
| **StoryCloze** | EN 71.98 | ZH 60.29 | JA 59.06 | FR 62.69 | TH 57.35 | EN **73.45** | ZH 64.16 | JA 59.67 | FR 64.78 | TH 58.20 | EN 73.30 | ZH **65.25** | JA **62.00** | FR **66.25** | TH **60.14** |
| | TL 56.04 | DE 62.38 | RU 61.53 | AR 54.41 | | TL 56.58 | DE 64.09 | RU 64.40 | AR 56.50 | | TL **57.04** | DE **65.94** | RU **65.40** | AR **58.36** | |
| **XWinograd** | EN 73.12 | ZH 54.96 | JA 54.01 | FR 55.42 | RU 54.29 | EN 78.37 | ZH 58.93 | JA 54.01 | FR 57.83 | RU 53.65 | EN **78.97** | ZH **59.92** | JA **55.16** | FR **59.03** | RU **58.41** |
| **Flores** | | ZH 8.08 | JA 8.40 | FR 18.54 | TH 10.56 | | ZH 10.30 | JA 9.54 | FR 24.81 | TH 12.53 | | ZH **11.05** | JA **9.93** | FR **29.89** | TH **14.41** |
| | TL 31.53 | DE 19.06 | RU 15.35 | AR 14.55 | | TL 35.15 | DE 23.12 | RU 17.24 | AR 17.17 | | TL **36.75** | DE **30.15** | RU **23.76** | AR **19.11** | |

Table 2: Comparison between ParaRater Stage 1 and the full two-stage ParaRater pipeline. Results are averaged across all evaluated languages.

| | ARC-E | BMLAMA | MMLU | HellaSwag | StoryCloze | XWinoGrad | Flores |
|---|---|---|---|---|---|---|---|
| Stage1 | **46.44** | **52.15** | 27.44 | 40.01 | 62.70 | 62.30 | 20.13 |
| Stage2 | 45.97 | 51.44 | **27.50** | **40.22** | **63.74** | 62.30 | **21.88** |

## 5.3 ABLATION STUDY: STAGE 1

We conduct an ablation study by removing Stage 2 to investigate the effect of using only Rater 1. Following the same experimental setup, we use Rater 1 to select 1B tokens per language, which are then combined with 100B English tokens for training. The result is shown in Table 2. We observe that Stage 2 significantly enhances performance on NLU and translation tasks over Stage 1. The improvements on StoryCloze and Flores are primarily attributed to Rater 2. Stage 1 mainly serves to filter out low-quality texts, thereby reducing noise in the overall distribution. Stage 2, in contrast, maintains the marginal distributions while perturbing the joint distribution between the two languages to identify the most strongly interacting parallel corpora. We also observe that on knowledge-centric tasks, Stage 1 shows a slight advantage over Stage 2. This finding aligns with our earlier attribution: since both raters are trained on the same validation set, Stage 2 tends to bias the rater further toward selecting data that benefits language understanding rather than knowledge acquisition. A promising direction for improvement is to explore more principled approaches for validation set selection.

Table 3: Samples classified according to ParaRater rankings in Stage 1 and Stage 2. In stage 1, ↑ and ↓ denotes high and low rank respectively. In Stage 2, ↑ denotes samples that keep the rank, whereas ↓ denotes samples whose ranks have declined.

| Category | English | Chinese |
|---|---|---|
| **Stage 1** | | |
| EN ↑ ZH ↑ | Amplifier circuits can be described in terms of the layout or topology of components which plays a crucial part in shaping the sound ... using a lead made from this between guitar and amp, while not dangerous, will usually be very noisy. | 放大器电路可以用元件的布局或拓扑来描述，这在塑造声音方面起着至关重要的作用。 大多数经典的电吉他放大器采用A类设计 ... 使用这种线材连接吉他和音箱，虽然不会有危险，但通常会很嘈杂。 |
| EN ↑ ZH ↓ | It's 1928 in Harlem, New York. Jazz and gangsters are king in an interesting lifestyle. Richard Gere plays Dixie Dwyer, a musician who witnesses a murder. ... | 1928年，纽约哈莱姆。 爵士乐和黑帮是一个有趣的生活方式的国王。 理查德·吉尔饰演迪克西·德怀尔，目击了一起谋杀案。 ... |
| EN ↓ ZH ↑ | The "Foothill Village" of Sierra Madre is a small, quaint and friendly town with storefronts reminiscent of the 1920s and 1930s, and is often thought of as an "artist' colony" ... | "塞拉玛德拉的"山麓村"是一个小，古朴而友好的小镇，店面令人联想到20世纪20年代和30年代，并经常被认为是一个"艺术家的殖民地"。超过40%的城镇的房屋超过50岁 ... |
| EN ↓ ZH ↓ | Materials: Chart paper with the riddle: ãShould Sheri share her shoes with her sister Shelly?ä on it. Class set of the book by Dr. Seuss, One Fish, Two Fish, Red Fish, Blue Fish. | 材料：一张纸上写着谜语："Sheri 应该和她的妹妹 Shelly 分享她的鞋吗？" Dr. Seuss 的书，《一只鱼，两只鱼，红鱼，蓝鱼》的班级套书。 |
| **Stage 2** | | |
| EN ↓ | People often message me on my blog with questions about why their cakes have gone wrong and how they can prevent it happening again. ... | 人们经常在我的博客上给我留言，问为什么他们的蛋糕出了问题，以及如何防止再次发生。... |
| EN ↑ | Over the past century, the neighborhood of Little Italy, Manhattan, has evolved from a cultural center for Italian immigrants to a tourist destination with very little Italian culture at all. ... | 在过去的一个世纪里，曼哈顿的小意大利区从意大利移民的文化中心演变为一个几乎没有意大利文化的旅游目的地 ... |

## 6 CASE STUDY AND DISCUSSION

We draw English-Chinese samples from the two-stage ParaRater scoring procedure to conduct a qualitative analysis of its scoring behavior.

In stage 1, the rater assigns scores to the bilingual corpus, which are then ranked separately within each language. We designate the top 30% as high-ranked and the bottom 30% as low-ranked, yielding four cross-lingual ranking combinations. The upper section of Table 3 illustrates representative cases from these four categories for Chinese–English data. We find that the four categories display clear patterns. In particular, documents ranked highly in both Chinese and English tend to be fluent, knowledge-bearing explanatory texts with relatively greater length. For documents that receive high rankings in English but low rankings in Chinese (**EN ↑ ZH ↓**), we observe that many instances, consistent with our earlier discussion, are attributable to poor or incorrect translations. In the case, "are king" is a figurative expression meaning "to dominate" or "to be the most important or influential thing." However, in Chinese this was translated literally as "king," which makes the sentence sound very confusing. Therefore, improving translation quality would reduce the number of samples falling into this category and potentially shift them toward the first category (**EN ↑ ZH ↑**). In the category where English ranks low but Chinese ranks high (**EN ↓ ZH ↑**), we observe that most samples pertain to topics such as Western geography, institutions, and culture. However, the domains of

these texts may be relatively distant from the distribution of the Chinese corpus, making the original English texts less effective in improving the model's Chinese capability. Nevertheless, these texts are generally fluent, and thus, once translated into Chinese, they may be recognized as high-quality Chinese data. For data that receive low scores in both languages (**EN ↓ ZH ↓**), we generally observe issues with fluency and completeness; the corresponding English texts are often disorganized and difficult to understand.

Stage 2 performs a decomposition of the subset that ranked highly in both languages (**EN ↑ ZH ↑**). In Table 3, we categorize samples whose rankings drop by more than 20% as having a significant decline (**EN ↓**); otherwise, they are considered to have maintained their ranking (**EN ↑**). In Stage 2, the samples that maintain high rankings (**EN ↑**) tend to be more well-structured and formal, with overall very fluent translations. In contrast, the samples whose rankings drop (**EN ↓**), although still high-quality in English, often have translations that appear somewhat stiff or unnatural. While more regular and polished text generally signals higher quality, it also reduces diversity to some extent. We take a reverse perspective to analyze the factors contributing to these translation quality outcomes. Fluently translated text is easier for the model to learn — after all, a translation model is itself an LLM. Consequently, the samples that drop in ranking in Stage 2 are harder to learn in the target language, and thus require the presence of parallel data to enable effective transfer.

Using ParaRater, we can either identify the core portion within existing parallel corpora or pre-select it on the source side to construct targeted parallel data, thereby maximizing the efficiency of corpus construction and cross-lingual transfer of linguistic competence. ParaRater learns language-specific selection strategies through meta-learning, offering a more adaptive and fine-grained alternative to existing methods that rely on fixed heuristics or rigid filtering criteria.

## 7 CONCLUSION

To address the challenge of parallel corpus selection and construction, this paper introduces ParaRater. By leveraging the scoring differences between the two raters, ParaRater identifies parallel data with strong cross-lingual interactions. ParaRater substantially improves the transfer efficiency of language abilities during LLM training. Extensive experiments are conducted from English to eight target languages to validate the approach. Results show that parallel data selected by ParaRater leads to significant gains in both NLU and translation performance, while maintaining stable performance on English tasks. An important direction for future work is to explore how to better choose and configure the training objectives for ParaRater.

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

# A  DETAILS OF META LEARNING PARARATER

Algorithm 1 outlines the rater training process, which is cast as a bi-level optimization problem. In the inner loop, the language model $f_\theta$ is updated on batches of English data from $\mathcal{D}_{\text{train}}$. For each batch, the rater $\phi_\eta$ assigns a score to every example, which is then converted into a non-negative weight. These weights determine how strongly each example influences the inner training loss $\mathcal{L}_{\text{inner}}$, encouraging the model to focus on English data that may be most valuable for transfer. In detail, we use the softmax function to get the normalized weights: $w(x) = \frac{e^{\phi_\eta(\mathbf{x})}}{\sum_{\mathbf{x}' \in B_t} e^{\phi_\eta(\mathbf{x}')}}$ .

In the outer loop, after $T$ such inner updates, the adapted model $f_{\theta^{(T)}}$ is evaluated on held-out target-language validation data $\mathcal{D}_{\text{val}}$. The validation loss $\mathcal{L}_{\text{val}}$ captures how well the model transfers knowledge across languages. By differentiating through the inner steps, we obtain meta-gradients with respect to $\eta$, which are used to update the rater. Over time, $\phi_\eta$ learns to assign higher weights to English examples whose translations improve target-language performance, while down-weighting less useful ones.

---
**Algorithm 1** Meta-Learning a ParaRater
---
**Inputs:** Train data $\mathcal{D}_{\text{train}}$, validation data $\mathcal{D}_{\text{val}}$; inner model $f_\theta$; rater $\phi_\eta$ producing per-sample scores/weights; inner steps $T$, outer steps $K$.

  1: Initialize $\theta, \eta$
  2: **for** $k = 1$ to $K$ **do**                                  ▷ Outer loop (meta step)
  3:     $\theta^{(0)} \leftarrow \text{CLONE}(\theta)$
  4:     **for** $t = 1$ to $T$ **do**                          ▷ Inner loop (train LM with rater weights)
  5:         Sample batch $B \subset \mathcal{D}_{\text{train}}$
  6:         $s \leftarrow \phi_\eta(B)$                              ▷ Rater scores
  7:         $w \leftarrow \text{SCORESTOWEIGHTS}(s)$          ▷ Any monotone map/normalization
  8:         $\mathcal{L}_{\text{inner}} \leftarrow \sum_{(x,y) \in B} w(x) \cdot \text{CE}(f_{\theta^{(t-1)}}(x), y)$
  9:         $\theta^{(t)} \leftarrow \theta^{(t-1)} - \alpha \nabla_{\theta^{(t-1)}} \mathcal{L}_{\text{inner}}$
 10:     **end for**
 11:     Sample validation batch $\tilde{B} \subset \mathcal{D}_{\text{val}}$
 12:     $\mathcal{L}_{\text{val}} \leftarrow \frac{1}{|\tilde{B}|} \sum_{(\tilde{x},\tilde{y}) \in \tilde{B}} \text{CE}(f_{\theta^{(T)}}(\tilde{x}), \tilde{y})$
 13:                                                                      ▷ Meta-update
 14:     $\eta \leftarrow \eta - \beta \nabla_\eta \mathcal{L}_{\text{val}}\left(\theta^{(T)}(\eta)\right)$
 15:     $\theta \leftarrow \text{COPY}(\theta^{(T)})$
 16: **end for**
 17: **return** $\phi_\eta$                                           ▷ Trained rater
---

To further ensure the quality of selected data, ParaRater employs a two-stage filtering strategy. First, it selects examples that are helpful on their own when weighted by the rater. Second, it retains only those that prove effective when both the English and translated versions are included in training, thus capturing truly impactful parallel corpora. The final selected set can then be translated to build high-quality parallel corpora, enabling more efficient and effective cross-lingual transfer for multilingual LLMs.

# B  EXPERIMENTAL DETAILS

The hyper-parameter settings employed for the meta-learning of ParaRater are listed below.

For the pretraining experiments, we employ a transformer architecture derived from the LLaMA-2 model (Touvron et al., 2023b), configured with roughly 1.2 billion parameters. All models are initialized randomly prior to pretraining. Table 4 provides the full specifications of the model architecture and training hyperparameters.

For rater training, we initialize with Qwen3-Embedding-0.6B and using a inner model with the same architecture of the language model as in the pretraining. During the meta-learning process, we set

| Model configuration | Values |
|---|---|
| Attention head | 16 |
| Layers | 24 |
| Hiddent size | 2048 |
| Intermediate layer dimension | 5504 |
| maximum position embedding | 4096 |
| layer normalization epsilon | $1 \times 10^{-5}$ |
| **Training Hyperparameters** | **Values** |
| Batch size | 3072 |
| Sequence length | 4096 |
| Optimizer | AdamW |
| Learning rate | $4.3 \times 10^{-4}$ |
| Learning rate schedule | Cosine decay to 10% of inital value |
| Traning steps | Varied based on the total token budget |
| Precision | bf16(mxied-precision training) |

Table 4: Model configuration and Training Hyperparameters for pretraining LLms

the number of inner steps to 2, and the maximum token length for both the rater and the inner model is fixed at 512.

