# OpenReview forum: "ParaRater: Enhancing Cross-Lingual Transfer in Large Langauge Models with Meta-Learning"
_ICLR.cc/2026/Conference — ICLR 2026 Conference Withdrawn Submission_

### Official Review · Reviewer_4mvy · 2025-10-24

**Soundness:** 2
**Presentation:** 2
**Contribution:** 3
**Rating:** 8
**Confidence:** 2

**Summary:**

The central question of the paper is:
Given a pair of parallel documents (e.g. English original and Chinese translation),
how much does it contribute to the multilingual performance of a language model, when included into the pretraining data?
More specifically, the authors care about cases in which the *combination* of the two documents drives performance, not either language version on its own.

To address this question, they introduce a meta-learning approach/model called *ParaRater*.
In a first stage, ParaRater is trained to rate each document independently (i.e., ignoring parallelism).
Pairs of documents that get a high score in both languages are retained.
In a second stage, a new instance of ParaRater is trained on a subset of the corpus:
source-language documents that got a high score, plus *mismatched* target-language documents that also got a high score, plus some low-scoring source-language documents.
The authors retain only those source-language documents (and corresponding target-language versions) whose ranking dropped in the second stage,
arguing that these documents only contribute because of parallelism.

**Strengths:**

**Originality:**
The paper introduces a new approach to rate documents in parallel corpora for their contribution to quality of multilingual language models.
In particular, it disentangles truly parallel document contributions from those in which only the source-language or only the target-language version contributes to model performance.

**Quality:**
The approach appears methodologically sound.

**Clarity:**
Fair enough, but could be improved (see below).

**Significance:**
The contribution could lead to a new way of assessing the importance of parallel documents for multilingual LM pretraining.
The authors mention one potential use case that seems particularly exciting:
assessing the importance of source-language (usually English) documents *before* they are translated,
thus guiding translation efforts.

**Weaknesses:**

**"Low-resource":**
You should clarify what you mean by "low-resource languages".
There are several definitions in the literature (e.g. https://aclanthology.org/2024.emnlp-main.983.pdf , https://aclanthology.org/2020.acl-main.560.pdf).
Some of the target languages you are investigating are actually very high-resource according to most definitions (Chinese, French, German);
at best they are lower-resource compared to English.
This is a real limitation:
You are implicitly assuming that machine translation into the target languages is possible (if expensive).
For truly low-resource languages, a human translation effort would be required, which is of course even more expensive.
This limitation is worsened by the fact that you train a separate ParaRater for each language.
So if you want to advertise your work with the "low-resource" keyword, you should at least acknowledge this limitation.

Some of the open-ended questions in my next answer may lead towards addressing this limitation.

**Clarity:**
Especially section 4 is hard to follow:
It is difficult to understand the structure of the section at a glance.
Relatedly, figure 1 is also hard to follow.
To improve clarity, you could in particular:

* announce the structure at the beginning of the section, and in particular describe each of the two stages in ~1 sentence;
* provide a more intuitive description of stage 2, in addition to the formal description you already give;
* state more prominently that the rater scores source and target language documents separately;
* state more prominently that you are training a separate ParaRater for each target language;
* boldface *ParaTool* and *ParaCore* on lines 332-224, since your introduction mentions them as separate contributions;
* answer the clarification questions below.

**Grammatical errors:**
* l. 192: "a high-value pre-determined size of subset" -> "a high-value subset of pre-determined size"
* l. 307 "other data select method" -> "other data selection methods"

**Questions:**

Clarification questions:

* l. 290-292: "*Unlike some previous work, we do not use concatenated parallel pairs in order to eliminate potential bias introduced by the data format; instead, all parallel
documents are distributed independently within the training data.*"
Which previous work? And why do you do this?
* algorithm 1 (l. 717-738): Is $\theta^{(0)}$ the same at every step of the outer loop, or is it the old $\theta^{(t)}$? At present this is unclear to me.

More open-ended / future-work questions:
* l. 273-283 (source-only selection): Is this just a hypothetical scenario or have you actually tried something of the sort? For source-only selection, would you reuse a ParaRater you trained on a parallel corpus, or retrain a new one? If you train a new one, how would you adapt the training recipe?
You should say more about this, since this is possibly the most important use case.
* Does a ParaRater trained on one LM architecture/size transfer to another LM architecture/size?
* Do ParaRaters trained for different target languages select different source language (English) documents? What are the differences, if any?
* Depending on the answer to the previous question: Have you considered training just one ParaRater for a set of languages (instead of a separate one for each language)?

---

> ### Author Response · Authors · 2025-11-21
>
> Thank you for your thoughtful review and feedback on our submission. We appreciate the time and effort you invested in understanding our work. We would like to address the concerns you raised:
>
> > **Q1:** "Low-resource": You should clarify what you mean by "low-resource languages".
> >
>
> **A1:**
>
> Thank you very much for your careful consideration and for pointing this out. Our paper focuses on **cross-lingual transfer**, which in practice is almost equivalent to studying how to transfer **English capabilities** to other languages. Therefore, first, all non-English languages are treated as low-resource relative to English. In addition, **when we explicitly use the term *low-resource* in the paper, it primarily refers to languages like Tagalog, where native corpora are comparatively scarce.** Thus, **we are not emphasizing extremely rare languages**—those that current powerful LLMs are not even capable of supporting. Our method indeed requires machine translation to be feasible.
>
> Thank you very much for pointing this out, we will clarify this more explicitly in the paper.
>
> > **Q2:** Clarity: Especially section 4 is hard to follow: It is difficult to understand the structure of the section at a glance. Relatedly, figure 1 is also hard to follow.
> >
>
> **A2:**
>
> Thank you for your valuable suggestions. From your comments, we can clearly see that you have developed a deep understanding of our work, and we truly appreciate the time and effort you devoted to reviewing our paper. We also noticed that other reviewers encountered some misunderstandings regarding Section 4, so we will take your suggestions very seriously and improve the clarity of our presentation accordingly.
>
> > **Q3:** l. 290-292: "Unlike some previous work, we do not use concatenated parallel pairs in order to eliminate potential bias introduced by the data format; instead, all parallel documents are distributed independently within the training data." Which previous work? And why do you do this?
> >
>
> **A3:**
>
> What we intend to express here is that, in some prior work on parallel data **such as [1]**, the parallel sentences are concatenated in a specific format. In contrast, we do not adopt such concatenation; instead, we simply mix the parallel documents into the training data. Our decision is based on earlier experiments showing that the **formatting of parallel data has little impact on a model’s general multilingual ability, and mainly affects translation performance**. This is because most concatenation formats resemble translation-task prompts, which effectively function as SFT for translation. Paper [2] has the same observation as ours, which you might be interested.
>
> [1] Cheng, Shanbo, et al. Seed-x: Building strong multilingual translation llm with 7b parameters. *arXiv preprint arXiv:2507.13618* (2025).
>
> [2] Anonymous Authors. From Translation to Multilinguality: Revisit the Role of Parallel Data in Multilingual LLM Pretraining. 2025. Under review.
>
> > **Q4:** algorithm 1 (l. 717-738): Is the same at every step of the outer loop, or is it the old ? At present this is unclear to me.
> >
>
> **A4:**
>
> They are not the same. As you can see in line 15 in Alg. 1, at the end of each outer-loop iteration, we save the current inner language model parameters $\theta$, and at the beginning of the next outer loop, we assign this value to $\theta^{(0)}$. Therefore, the initial inner language model parameters for each outer loop are based on those from the previous iteration. We write it this way to distinguish this from the inner-loop updates, which use the notation $\theta^{(t)}$, where t denotes the inner steps.
>
> > **Q5:** l. 273-283 (source-only selection): Is this just a hypothetical scenario or have you actually tried something of the sort? For source-only selection, would you reuse a ParaRater you trained on a parallel corpus, or retrain a new one? If you train a new one, how would you adapt the training recipe? You should say more about this, since this is possibly the most important use case.
> >
>
> **A5:**
>
> **We only pre-translate and construct parallel data during the training of the rater.** **Once the rater is trained, it no longer relies on bilingual data for selection.**
>
> Yes, source-only selection is the most important use case. **In Section 5 (our main experiments), ParaRater is evaluated entirely under the single-sided selection setting**: we first select English documents using the learned rater, and only then translate the selected small subset, **thus assuming no availability of large-scale parallel corpora**. This does not require retraining the rater.

---

> > ### Author Response · Authors · 2025-11-21
> >
> > > **Q6:** Does a ParaRater trained on one LM architecture/size transfer to another LM architecture/size?
> > >
> >
> > **A6:**
> >
> > **Pretraining experiments require substantial computational resources**, and given our current limitations, we have not yet results on larger models (e.g., 7B). We are also actively conducting larger-scale experiments.
> >
> > Based on existing empirical evidence, we believe that extending **the conclusions observed at the 1.2B scale to a 7B model is quite promising**. We note that many studies [1,2] use small models to estimate the effectiveness of data selection on performance trends of larger models—some even use models with only 1M parameters [1]. This reflects the strong structural commonality within Transformer models.
> >
> > Following recent experimental practices in research on training data [3,4], our pretraining experiments at the 1.2B scale are among the large settings. This scale is sufficient to draw convincing conclusions.
> >
> > [1] Liu, Fengze, et al. "Quadmix: Quality-diversity balanced data selection for efficient llm pretraining." *arXiv preprint arXiv:2504.16511* (2025).
> >
> > [2] Guo, Ping, et al. "Exploring Polyglot Harmony: On Multilingual Data Allocation for Large Language Models Pretraining." *arXiv preprint arXiv:2509.15556* (2025).
> >
> > [3] Muhammad Reza Qorib, Junyi Li, and Hwee Tou Ng. 2025. Just Go Parallel: Improving the Multilingual Capabilities of Large Language Models. In *Proceedings of the 63rd Annual Meeting of the Association for Computational Linguistics (Volume 1: Long Papers)*, pages 33411–33424, Vienna, Austria. Association for Computational Linguistics.
> >
> > [4] Calian, Dan A., et al. "DataRater: Meta-Learned Dataset Curation." *arXiv preprint arXiv:2505.17895* (2025).
> >
> > > **Q7:** Do ParaRaters trained for different target languages select different source language (English) documents? What are the differences, if any?
> > >
> >
> > **A7:**
> >
> > The English data selected by ParaRater differs across target languages. This is precisely where ParaRater’s effectiveness becomes evident. Interestingly, we observe that languages exhibit different degrees of closeness in terms of the data they benefit from. For example, when using Japanese as the target language, about **11%** of the documents selected for Japanese also appear in the Chinese-selected set, **9%** in the Thai-selected set, and **7%** in the Tagalog-selected set, but only **3–4%** overlap with the sets selected for French, German, and Russian. This pattern reflects varying levels of content similarity or affinity among different languages.
> >
> > In future work, we will extend ParaRater to support more languages and further analyze the influence relationships among them.
> >
> > > **Q8:** Depending on the answer to the previous question: Have you considered training just one ParaRater for a set of languages (instead of a separate one for each language)?
> > >
> >
> > **A8:**
> >
> > Because prior work lacked a deep analysis of the impact of parallel data, earlier studies often mixed together the effects of the base model’s inherent capabilities, data distribution differences, and interactions among multiple languages. **Therefore, our goal in this paper is to establish a foundational analysis by first conducting a clean and controlled comparison in a bilingual setting.**
> >
> > Building on this foundation, **merging the per-language raters into a unified multilingual rater** will be a key direction for our future work. There are two possible approaches:
> >
> > 1. **Model distillation.**
> >
> >     We can treat the individual language-specific raters as teachers and train a single student model to learn their scoring strategies. This approach not only consolidates the existing raters into one unified model, but also has the potential to generalize to additional languages (e.g., languages within the same family).
> >
> > 2. **Direct multilingual training.**
> >
> >     Alternatively, we can follow the current ParaRater training pipeline, mix parallel data from multiple languages, and directly train a multilingual rater. This approach can capture not only the influence of English on other languages, but also the interactions *between* non-English languages, potentially leading to an even more optimal scoring strategy in multilingual settings.
> >
> >
> > There remain many promising directions for further improving the effectiveness of parallel data in cross-lingual transfer. ParaRater provides a foundational analysis and validation toward this goal. **We have also released the training code, the selected parallel data, and all rater models to support and encourage continued research within the multilingual community.**
> >
> > Thank you very much for the time you devoted to reviewing our paper. The questions you raised clearly reflect deep insight into the relevant research area, and we would be very happy to continue the discussion with you.

---

> > > ### Comment · Reviewer_4mvy · 2025-11-25
> > > **how realistic is the envisioned scenario?**
> > >
> > > Thank you for your answer!
> > >
> > > I will keep my score because I still think the paper introduces an interesting novel
> > > idea.
> > >
> > > However, I agree with the other reviewers that a better argument is needed for why source-side selection is a realistic scenario, given the need for machine translation. Based on your answers, your goal seems to be to reduce the amount of data you need to machine-translate.
> > >
> > > This prompted me to look at the numbers more closely: you are using 20B English tokens plus translation to train a ParaRater, and then ParaRater chooses 1B out of 100B English tokens to translate and add to LM pretraining data. But then, why not directly add the original 20B tokens plus translations to the LM pretraining data?

---

> > > > ### Author Response · Authors · 2025-11-25
> > > >
> > > > Thank you for acknowledging our work. Regarding the new concern you raised, our response is as follows:
> > > >
> > > > > **Q9:** I agree with the other reviewers that a better argument is needed for why source-side selection is a realistic scenario
> > > > >
> > > >
> > > > **A9:**
> > > >
> > > > We understand that the other reviewers’ concerns are not about if single-sided selection is a realistic scenario, but rather about a misunderstanding of our experiments—they may not realize that our experiments were conducted under the single-sided selection setting (**Reviewer fq6e Q5**).
> > > >
> > > > Using machine-translated parallel corpora is a widely adopted practice for pretraining multilingual LLMs and improving multilingual capability [1,2,3]. However, previous approaches are empirical, without any deeper analysis of which parts of the parallel data are actually valuable. In this context, providing a theoretical basis for single-sided selection is of significant practical importance.
> > > >
> > > > [1] Cheng, Shanbo, et al. "Seed-x: Building strong multilingual translation llm with 7b parameters." *arXiv preprint arXiv:2507.13618* (2025).
> > > >
> > > > [2] Dou, Longxu, et al. "Sailor2: Sailing in South-East Asia with Inclusive Multilingual LLMs." *arXiv preprint arXiv:2502.12982* (2025).
> > > >
> > > > [3] Üstün, Ahmet, et al. "Aya model: An instruction finetuned open-access multilingual language model." *Proceedings of the 62nd Annual Meeting of the Association for Computational Linguistics (Volume 1: Long Papers)*. 2024.
> > > >
> > > > > **Q10:** Based on your answers, your goal seems to be to reduce the amount of data you need to machine-translate.
> > > > >
> > > >
> > > > **A10:**
> > > >
> > > > This is one of the benefits brought by ParaRater. With ParaRater’s guidance, we can significantly reduce the amount of translation required and **allocate a larger portion of the training data budget to other sources.** This is especially important when parallel data is used for many languages: **indiscriminately generating and using parallel data not only increases cost but also reduces the proportion of other useful data in the training corpus.**
> > > >
> > > > ParaRater, by contrast, aligns the construction of parallel data with the model’s capability needs. Our work is the first to move **from empirical heuristics to a theory-driven understanding** of how parallel data should be synthesized and used.
> > > >
> > > > > **Q11:** But then, why not directly add the original 20B tokens plus translations to the LM pretraining data?
> > > > >
> > > >
> > > > **A11:**
> > > >
> > > > As mentioned above, using too much parallel data consumes the quota that could otherwise be allocated to other types of data. In our experimental setup, if we were to include all parallel data, the parallel portion alone would amount to **20 B × 8 tokens**, which would **drastically reduce the proportion of English data and significantly increase the total data volume**. Moreover, **much of this crude parallel data may not be useful, and some of it can even be harmful.**
> > > >
> > > > The 20B English tokens we translate are used only for training ParaRater, and this is extremely small compared to the **total amount of English data (15T in FineWeb)**. It is not feasible to translate the entire English corpus, so source-side selection is essential.
> > > >
> > > > We hope our responses address your concerns, and we welcome any further questions you may have.

---

### Official Review · Reviewer_W5xU · 2025-10-27

**Soundness:** 2
**Presentation:** 2
**Contribution:** 2
**Rating:** 2
**Confidence:** 4

**Summary:**

This work focuses on selecting parallel data used for pre-training of LLMs for facilitating their cross-lingual transfer capabilities.
Given a set of corpora, a two-stage filtering strategy is applied.
First,

**Strengths:**

- The proposed method is simple, and the experiments are extensive, including those on pre-training.
- The case study examines intuitive examples, providing useful insights for future research on cross-lingual transfer.
- To the best of my knowledge, there has been little work focusing on data selection for pre-training large language models, particularly with regard to enhancing cross-lingual transfer capabilities.

**Weaknesses:**

- I believe the overall presentation could be improved. For instance, the notations surrounding Equation (5) are somewhat complex and could be simplified for clarity. Figure 1 may also be enhanced to more effectively illustrate the concept—particularly the Stage 2 component, which is difficult to interpret in its current form. Moreover, the paper lacks sufficient details on how the selected data are actually utilized. If the data are used merely for language modeling, the role of parallel data remains unclear. Since this work focuses on parallel data, it would be worthwhile to explore alternative ways of leveraging such data beyond conventional language modeling, which can also be performed using monolingual data. The unique merits of parallel data can—and should—be investigated further, rather than using them simply as another component of standard autoregressive pre-training.
- The performance of the proposed method is not sufficiently convincing. Although the baselines employed are rather general methods for filtering pre-training data and do not explicitly address the research question of enhancing cross-lingual transfer, they generally outperform the proposed approach. It would strengthen the paper to include ablation studies that analyze key algorithmic choices. For example, why was Qwen3-Embedding-0.6B selected as the initial rater? Providing justification and comparative evidence for such decisions would improve the paper’s reliability.
- I also have concerns regarding the practical feasibility of the proposed method. The work centers on selecting parallel data, yet the experiments rely on machine-translated data due to the scarcity of high-quality parallel corpora. While parallel data can be beneficial, in practice, obtaining such data at scale and with sufficient quality is challenging. Given this limitation, it may be more practical to explore how to improve cross-lingual transfer using non-aligned, monolingual data instead.

**Questions:**

- Typo (L139): curpus -> corpus

---

> ### Author Response · Authors · 2025-11-21
>
> Thank you for your thoughtful review and feedback on our submission. We appreciate the time and effort you invested in understanding our work. We would like to address the concerns you raised:
>
> > **Q1:** I believe the overall presentation could be improved. For instance, the notations surrounding Equation (5) are somewhat complex and could be simplified for clarity. Figure 1 may also be enhanced to more effectively illustrate the concept—particularly the Stage 2 component, which is difficult to interpret in its current form.
> >
>
> **A1:**
>
> When reading Figure 1, you can focus on the trajectory of **Text 1**, which represents a document that receives high scores in both Chinese and English. After Stage 1, Text 1 is selected from the parallel data through Rater 1 . We then remove the Chinese side of Text 1 and keep only its English version. In Stage 2, Rater 2 scores the data again, and we observe whether the ranking of Text 1 drops. This corresponds to the two arrowed outcomes shown in the figure. If Text 1 maintains its ranking, this indicates that it does not rely on the Chinese version. In contrast, if its ranking drops significantly, this means that it requires both the English and Chinese versions to be present in order to be effective, and therefore it is identified as part of the *core of parallel data*.
>
> For Eq.5, we would like to clarify it for you here. First, $\mathbf{d}_s$ denotes the subset selected from the original English corpus, and $\mathbf{d}_t$ refers to its translated counterpart in the target language. In addition:
>
> - A tilde ( $\sim$ ) indicates the subset selected by **Rater 1**.
> - An asterisk ( ∗ ) indicates the subset selected by **Rater 2**.
> - A prime ( ' ) denotes the **mismatch set** that does *not* form a parallel pair with $\mathbf{d}_s$.
>
> **In Section 4.1, we first define the task of parallel data selection and the final optimization objective (Eq. 4).** This objective contains two components. The first requires the selected parallel data to minimize the model’s loss on the validation set. The second is a constraint that ensures the selected subset leads to maximum language-model loss when its target-language counterpart is not present. This constraint is what guarantees the selection of the core of parallel data.
>
> **However, jointly optimizing these two components involves discrete sampling and is not computationally feasible.** Therefore, in Section 4.2, we explain how we approximate this objective using **dual-rater cross filtering**. Stage 1 approximates the first component of Eq. 4, and Stage 2 further filters the subset obtained in Stage 1 to satisfy the second constraint in Eq. 4.
>
> Thank you for your suggestion. We will improve the wording and the figures in the paper to make the workflow easier to follow.

---

> > ### Author Response · Authors · 2025-11-21
> >
> > > **Q2:** Moreover, the paper lacks sufficient details on how the selected data are actually utilized. If the data are used merely for language modeling, the role of parallel data remains unclear. Since this work focuses on parallel data, it would be worthwhile to explore alternative ways of leveraging such data beyond conventional language modeling, which can also be performed using monolingual data. The unique merits of parallel data can—and should—be investigated further, rather than using them simply as another component of standard autoregressive pre-training.
> > >
> >
> > **A2:**
> >
> > We use the selected data in a very straightforward way as elaborated in the experiment setup: we add it to the training corpus to help improve the model’s multilingual performance.
> >
> > **Using parallel data to transfer English capabilities to other languages is an important approach for improving a model’s multilingual ability.** Many prior works have explored this direction [1,2,3,4]. However, as we noted in the paper, prior works lack an in-depth analysis of the impact of parallel data. Therefore, our focus is to analyze **what kinds of parallel data actually enhance a model’s multilingual ability**, and to explore **how such data can be selected and constructed effectively**. Moreover, since **the pretraining stage is the foundation of a model’s multilingual capability**, and because **research on the role of parallel data during this stage is almost nonexistent**, **we focus on pretraining stage**. You mentioned, “exploring alternative ways of leveraging such data beyond conventional language modeling,” might be a direction for parallel data application but out of our current scope.
> >
> > We would also very much appreciate it if you could share the specific works you have in mind when referring to “alternative ways.”
> >
> > [1] Yingli Shen, Wen Lai, Shuo Wang, Ge Gao, Kangyang Luo, Alexander Fraser, and Maosong Sun. 2025. From Unaligned to Aligned: Scaling Multilingual LLMs with Multi-Way Parallel Corpora. *EMNLP 2025*.
> >
> > [2] Fujii, K., Nakamura, T., Loem, M., Iida, H., Ohi, M., Hattori, K., ... & Okazaki, N. 2024. Continual pre-training for cross-lingual llm adaptation: Enhancing japanese language capabilities. *arXiv preprint arXiv:2404.17790*.
> >
> > [3] Hongyuan Lu, Haoyang Huang, Shuming Ma, Dongdong Zhang, Wai Lam, Zhaochuan Gao, Anthony Aue, Arul Menezes, and Furu Wei. 2023. TRIP: Accelerating Document-level Multilingual Pre-training via Triangular Document-level Pre-training on Parallel Data Triplets. *EMNLP Findings 2023.*
> >
> > [4] Muhammad Reza Qorib, Junyi Li, and Hwee Tou Ng. 2025. Just Go Parallel: Improving the Multilingual Capabilities of Large Language Models. *ACL 2025*.
> >
> > > **Q3:** The performance of the proposed method is not sufficiently convincing. Although the baselines employed are rather general methods for filtering pre-training data and do not explicitly address the research question of enhancing cross-lingual transfer, they generally outperform the proposed approach. It would strengthen the paper to include ablation studies that analyze key algorithmic choices. For example, why was Qwen3-Embedding-0.6B selected as the initial rater? Providing justification and comparative evidence for such decisions would improve the paper’s reliability.
> > >
> >
> > **A3:**
> >
> > We conduct strictly fair comparative experiments across eight languages spanning high-, mid-, and low-resource settings as well as multiple language families. The results (Table 1) show that **ParaRater brings substantial improvements to the model’s multilingual capabilities compared to previous data selection methods**. As you pointed out, we are the first to propose a data-selection approach specifically aimed at enhancing **cross-lingual transfer**.
> >
> > For the choice of the initial rater model, we also experimented with **bert-base-multilingual-cased**. After training both rater models in Stage 1, we applied them to the same data (50B English tokens) and compared the top 10% selected by each model. We found that the selected documents overlapped by **96%**. This indicates that our method is **not sensitive** to the choice of the rater model, and that even a smaller rater model can effectively perform the selection task.
> >
> > **We have released all training and data-selection code, the trained Raters, and the selected parallel datasets.** **We sincerely hope that you will consider the contribution this work makes to the multilingual research community.**

---

> > > ### Author Response · Authors · 2025-11-21
> > >
> > > > **Q4:** I also have concerns regarding the practical feasibility of the proposed method. The work centers on selecting parallel data, yet the experiments rely on machine-translated data due to the scarcity of high-quality parallel corpora. While parallel data can be beneficial, in practice, obtaining such data at scale and with sufficient quality is challenging. Given this limitation, it may be more practical to explore how to improve cross-lingual transfer using non-aligned, monolingual data instead.
> > > >
> > >
> > > **A4:**
> > >
> > > We would like to clarify a potential misunderstanding of our method.
> > >
> > > **ParaRater does not require a large amount of parallel data; instead, it improves the data efficiency of synthetic parallel corpora.**
> > >
> > > During rater training, we translate an mount of English documents to obtain their target-language versions $\mathbf{d}_t$. This step is **necessary** because optimizing the rater parameters requires access to both the English document $\mathbf{d}_s$ and its target-language translation $\mathbf{d}_t$ in order to assess the quality of a potential parallel pair. **Once the rater is trained, we can perform single-sided selection directly on the English corpus, rather than relying on a large pre-translated parallel corpus.**
> > >
> > > **In Section 5 (our main experiments), ParaRater is evaluated entirely under the single-sided selection setting**: we first select English documents using the learned rater, and only then translate the selected small subset, **thus assuming *no availability* of large-scale parallel corpora**.
> > >
> > > Additionally, we also understand that your concern questions the use of parallel data itself. We would like to provide more context on this point. **Using parallel data for cross-lingual capability transfer has become a key practice in training multilingual large language models**. Many open-source models that emphasize multilingual performance report the use of machine-translated parallel data in their technical documentation [1,2,3]. Our work is built upon this broader context and, for the first time, provides solutions for analyzing the properties of parallel data and for constructing such data efficiently.
> > >
> > > [1] Cheng, Shanbo, et al. "Seed-x: Building strong multilingual translation llm with 7b parameters." *arXiv preprint arXiv:2507.13618* (2025).
> > >
> > > [2] Dou, Longxu, et al. "Sailor2: Sailing in South-East Asia with Inclusive Multilingual LLMs." *arXiv preprint arXiv:2502.12982* (2025).
> > >
> > > [3] Üstün, Ahmet, et al. "Aya model: An instruction finetuned open-access multilingual language model." *Proceedings of the 62nd Annual Meeting of the Association for Computational Linguistics (Volume 1: Long Papers)*. 2024.
> > >
> > > We hope our clarifications address your concerns and answer your questions. Thank you for your valuable feedback. We sincerely hope that you will consider the novelty of our work and the contribution it makes to the multilingual research community.

---

### Official Review · Reviewer_fq6e · 2025-10-31

**Soundness:** 1
**Presentation:** 2
**Contribution:** 1
**Rating:** 2
**Confidence:** 4

**Summary:**

The study is motivated by the multilingual performance gap in LLMs. Investigate which types of parallel corpora contribute most to cross-lingual transfer performance by introducing ParaRater which is aimed at supporting corpus selection by identifying the most relevant English text to translate. This selection is adaptive based on the target language by leveraging meta learning. The authors apply their framework to create the ParaCore training dataset

**Strengths:**

- While meta-learning for data selection is well established, the authors introduce novel approach specific for task-specific fine-tuning for multilingual models.
- The idea of selecting corpora to build task or language specific is relevant and interesting.
- The study provides good illustrative examples on how this framework can be used.

**Weaknesses:**

- Mismatch between the framework explained in the introduction and the framework presented in the abstract and discussion. Filtering source and target language corpora requires access to parallel data, which conflicts with the proposed benefit of selecting source language data to translate.
- Missing baseline: The proposed data selection approach should be compared to a random data selection from the same corpus (FineWeb 2).
- The corpus selection is applied to the highly curated fineweb 2 corpus. To assess the effectiveness of ParaRater, the method should also be applied to more noisy, non-curated web corpora.
- Parts of the description of the proposed ParaRater method are unclear, introducing multiple adaptations and changes to the objectives after initial descriptions
  - assumes D_{test} is available in the source language
- The only realistic experimental setup is described in lines 279-283. As the authors correctly note, all previous methodologies dependent on test data (or machine translated test data), which is unrealistic.
- COMET based filtering makes the experimental setup more unrealistic, as the results are biased towards easy to translate texts

**Questions:**

- Please clarify how you envision the framework applied in a practical setting without access to abundant parallel corpora. Please also specify which results presented in the manuscript correspond to this setting.

- typo line 139

---

> ### Author Response · Authors · 2025-11-21
>
> Thank you for your thoughtful review and feedback on our submission. We appreciate the time and effort you invested in understanding our work. We would like to address the concerns you raised:
>
> > **Q1:** Mismatch between the framework explained in the introduction and the framework presented in the abstract and discussion. Filtering source and target language corpora requires access to parallel data, which conflicts with the proposed benefit of selecting source language data to translate.
> >
>
> **A1:**
>
> You may have misunderstood the training and usage pipeline of ParaRater. During training, we use parallel data constructed through pre-translation as the training input. **After the Rater is trained, there are two application scenarios:**
>
> 1. **Double-sided selection: When parallel data is available**, ParaRater can directly select high-quality parallel samples.
> 2. **Single-sided selection: When only monolingual data is available**, ParaRater selects the most valuable monolingual samples, which are then translated into target languages to form parallel data.
>
> Since naturally occurring parallel corpora are extremely scarce, scenario 2 is the more practical and more valuable setting in real applications. Therefore, **our experiments focus on validating ParaRater under scenario 2**. Specifically, we apply ParaRater only on English monolingual data, and then translate the selected samples into 8 target languages.
>
> Figure 1 illustrates the training process of ParaRater. We understand that it may appear somewhat complex, and we will improve both the figure and the accompanying explanation.
>
> > **Q2:** Missing baseline: The proposed data selection approach should be compared to a random data selection from the same corpus (FineWeb 2).
> >
>
> **A2:**
>
> We report the random selection results.
>
> The table below shows the averages across languages for each benchmark.
>
> | Benchmark | Random | ParaRater |
> | --- | --- | --- |
> | **ARC-E** | 44.39 | 45.97 |
> | **MMMLU** | 27.14 | 27.50 |
> | **BMLAMA** | 50.19 | 51.44 |
> | **HellaSwag** | 38.64 | 40.22 |
> | **StoryCloze** | 61.96 | 63.74 |
> | **XWinograd** | 51.45 | 62.30 |
> | **Flores** | 15.42 | 21.88 |
>
> **Random selection is a very weak baseline**, and this has been thoroughly demonstrated in many prior studies on data selection. Therefore, we chose two strong data selectors, DCLM and FineWeb-Edu, to compare with ParaRater. In our experiments, each selection method modifies only a relatively small portion of the data, 1B tokens per language, so the model performances remain relatively close. **If the proportion of parallel data were increased, the disadvantages of random selection would become much more pronounced.** Paper [1] also compares model performance when trained on FineWeb-Edu versus directly on the raw FineWeb2 dataset (Figures 10 and 11). As shown, using unfiltered data significantly degrades model performance.
>
> [1] Penedo, Guilherme, et al. "The fineweb datasets: Decanting the web for the finest text data at scale." *Advances in Neural Information Processing Systems* 37 (2024): 30811-30849.
>
> > **Q3:** The corpus selection is applied to the highly curated fineweb 2 corpus. To assess the effectiveness of ParaRater, the method should also be applied to more noisy, non-curated web corpora.
> >
>
> **A3:**
>
> **Though FineWeb2 improves upon earlier data-cleaning pipelines to some extent, it still contains a substantial amount of low-quality text (as can also be seen in our case examples).** Modern LLM pretraining almost never uses such raw data directly. It has become standard practice to apply additional filtering with classifiers such as DCLM and FineWeb-Edu to obtain higher-quality subsets.
>
> **We would also like to clarify that the goal of ParaRater is fundamentally different from existing data-selection methods**. Existing selectors such as DCLM and FineWeb-Edu aim to extract high-quality subsets directly from raw data. In contrast, ParaRater focuses on selecting data that is most beneficial for cross-lingual transfer. Therefore, **ParaRater is not a replacement for these methods**. In practice, it should be applied after general data-quality filtering as a further selection step. Although ParaRater naturally filters out low-quality text (as shown in our case study)—since low-quality English text is unlikely to contribute to effective cross-lingual transfer—this is not its primary function.
>
> **Our experiments are conducted under a fully fair setup:** we apply three different selection methods on the same English corpus to obtain equally sized subsets, translate them into multilingual parallel data, and train with the exact same number of tokens. Under these controlled conditions, we demonstrate ParaRater’s effectiveness.

---

> ### Author Response · Authors · 2025-11-21
>
> > **Q4:** Parts of the description of the proposed ParaRater method are unclear, introducing multiple adaptations and changes to the objectives after initial descriptions; assumes D_{test} is available in the source language
> >
>
> **A4:**
>
> We hope that the following explanation will help you understand the ideas behind ParaRater.
>
> **In Section 4.1, we first define the task of parallel data selection and the final optimization objective (Eq. 4).** This objective contains two components. The first requires the selected parallel data to minimize the model’s loss on the validation set. The second is a constraint that ensures the selected subset leads to *maximum* language-model loss when its target-language counterpart is not present. This constraint is what guarantees the selection of the *core of parallel* data.
>
> **However, jointly optimizing these two components involves discrete sampling and is not computationally feasible.** Therefore, in Section 4.2, we explain how we approximate this objective using **dual-rater cross filtering**. Stage 1 approximates the first component of Eq. 4, and Stage 2 further filters the subset obtained in Stage 1 to satisfy the second constraint in Eq. 4.
>
> **When training the rater via meta-learning, the evaluation set $D_{test}$ used to validate model performance does exist.** In our experiments, we use native corpora of the target languages sampled from FineWeb2 without any further processing. We use the model’s loss on these samples as a proxy for its ability in the target language.
>
> There is still substantial room for improving how $D_{test}$ is selected. Exploring strategies for choosing validation data that can further enhance model performance is an important direction for future research.
>
> > **Q5:** The only realistic experimental setup is described in lines 279-283. As the authors correctly note, all previous methodologies dependent on test data (or machine translated test data), which is unrealistic.
> >
>
> **A5:**
>
> There may be some misunderstanding. During the **training stage**, we use a small amount of pre-translated parallel data as training input so that the rater can learn how parallel data affects language-model performance. The evaluation set $D_{test}$ used to assess language-model performance consists of **native corpora in each target language**, and does not rely on translation.
>
> Once the rater has been trained, there are two typical application scenarios:
>
> 1. **Double-sided selection: When parallel data is available**, ParaRater can directly select high-quality parallel samples.
> 2. **Single-sided selection: When only monolingual data is available**, ParaRater selects the most valuable monolingual samples, which are then translated into target languages to form parallel data.
>
> The second scenario is more valuable and is the primary goal of our method. Therefore, **all of our experiments are conducted under this single-sided selection** setting. In lines 279–283, we describe the difference between using ParaRater for **single-sided selection** and **double-sided selection**.
>
> > **Q6:** COMET based filtering makes the experimental setup more unrealistic, as the results are biased towards easy to translate texts
> >
>
> **A6:**
>
> On the contrary, **it is precisely the use of COMET for controlling translation quality that prevents the rater from being biased toward selecting samples that are simply easier to translate.** When constructing **the training data** for the rater, we first pre-translate a batch of parallel data and then filter out low-quality translations using COMET. This step is designed to improve the **efficiency and quality** of the training data. If the training set contains too many poorly translated pairs, the useful signal becomes diluted, and the overall training quality degrades. Moreover, when translation quality varies widely within the data, the rater may overfit to superficial cues and end up selecting samples that are merely easy to translate.
>
> **In the Section 5 experiments, where we use ParaRater selects data on the source side, we do not use COMET for any additional filtering.** As shown in our case study, ParaRater does **not** simply select fluent or easy-to-translate text.

---

> > ### Author Response · Authors · 2025-11-21
> >
> > > **Q7:** Please clarify how you envision the framework applied in a practical setting without access to abundant parallel corpora. Please also specify which results presented in the manuscript correspond to this setting.
> > >
> >
> > **A7:**
> >
> > We understand that your concerns are primarily centered around the **single-sided selection** setting. The answers provided above should already clarify most of these concerns.
> >
> > We construct a small amount of parallel data **during rater training**, which is necessary because the optimization of the rater’s parameters depends on both the selected English data $\mathbf{d}_s$ and its corresponding target-language version $\mathbf{d}_t$.
> >
> > However, **in the subsequent evaluation experiments (Section 5), we perform selection** **only on English**, and the method **does not rely on having large amounts of existing parallel data**.
> >
> > We hope our clarifications address your concerns and answer your questions. Once again, thank you for your valuable feedback. We sincerely hope that you will consider the novelty of our work and the contribution it makes to the multilingual research community.

---

### Official Review · Reviewer_NFEf · 2025-11-01

**Soundness:** 3
**Presentation:** 3
**Contribution:** 2
**Rating:** 4
**Confidence:** 4

**Summary:**

This paper addresses the lack of systematic understanding regarding which types of parallel corpora most effectively support cross-lingual transfer from English to other languages. The authors propose **ParaRater**, a meta-learning–based data selection method that identifies the most valuable English sentences to translate, by directly aligning corpus selection with model performance on native target-language data. Using a two-stage filtering mechanism, ParaRater isolates data instances that only improve performance when both their English and translated versions are included in training—capturing truly impactful parallel examples. Experiments across multiple languages demonstrate the method’s effectiveness.

**Strengths:**

1. The authors identify a highly valuable yet underexplored issue in multilingual that not all parallel data equally contribute to cross-lingual transfer.

2. Beyond merely selecting “more useful” sentence pairs, the work provides a deeper diagnostic by distinguishing whether performance gains stem from the **joint presence** of source and target sentences in training, or from English **independent contributions**. This insight helps isolate truly synergistic bilingual examples that drive effective cross-lingual knowledge transfer.

**Weaknesses:**

1. **Limited Methodological Novelty**:
   The distinction between the proposed method (ParaRater) and prior work such as **DataRater** is not clearly articulated. The approach appears to primarily adapt an existing framework to a new setting—cross-lingual transfer—by modifying the optimization objective: instead of merely selecting parallel examples with low loss, it favors those whose non-parallel counterparts yield significantly higher loss. While this shift is meaningful, the core technique lacks substantial innovation beyond the objective redesign.

2. **Missing Baseline Comparison**:
   The paper omits a critical baseline: **random selection of English sentences for translation**. Without this, it is difficult to assess whether the gains truly stem from intelligent data selection or simply from the inclusion of any additional parallel data.

3. **Scalability and Transferability Concerns**:
   It remains unclear whether the ParaRater selector must be retrained for different model architectures or scales (e.g., 7B vs. 70B LLMs). If a dedicated selector is required per model, the computational cost of training the selector may outweigh the marginal gains—especially when compared against the simpler random baseline.

4. **Evaluation Scope Limited to Pretraining**:
   The authors argue that parallel data is most effective during pretraining. However, given the prevalence of strong multilingual foundation models today, the community increasingly focuses on **post-training** (e.g., instruction tuning, alignment) for cross-lingual enhancement. The absence of experiments in post-training settings raises concerns about whether the observed pretraining benefits generalize to more practical, downstream adaptation scenarios.

5. **Lack of Scaling Analysis**:
   While the method is validated on a 1.2B-parameter model, its effectiveness on larger-scale LLMs (e.g., 7B, 13B, or beyond) is unexamined. As model capacity increases, the sensitivity to data quality may change—potentially diminishing the relative impact of curated parallel data. A scaling study would be essential to justify the method’s relevance in modern LLM development pipelines.

**Questions:**

See weakness.

---

> ### Author Response · Authors · 2025-11-21
>
> Thank you for your thoughtful review and feedback on our submission. We appreciate the time and effort you invested in understanding our work. We would like to address the concerns you raised:
>
> > **Q1:** Limited Methodological Novelty: The distinction between the proposed method (ParaRater) and prior work such as DataRater is not clearly articulated.
> >
>
> **A1:**
>
> **ParaRater and DataRater both use meta-learning for data selection, but they are fundamentally different.** DataRater selects the subset of the current data that is most beneficial for reducing the loss, whereas **ParaRater selects monolingual data that is most valuable to translate into other languages.** The goal of DataRater is to directly improve model performance on the validation set, while the goal of ParaRater is to maximize data efficiency for cross-lingual transfer.
>
> **Our approach begins with the problem, and then identifies and adapts appropriate techniques rather than pursuing innovation merely for the sake of being novel. Our motivation is to understand what kinds of parallel data are useful, and how to construct such parallel corpora through selection and translation**—an aspect that has never been explored before. Starting from this motivation, we define the concept of the core of parallel data and propose a meta-learning–based method for building parallel corpora.
>
> Our innovations include:
>
> - **Defining the concept of *core of parallel data*** and, for the first time, providing a decomposed analysis of the internal properties and effects of parallel corpora.
> - **Proposing a two-stage cross-filtering method** to address the high computational complexity of directly selecting core-of-parallel data.
> - **Our method supports both double-sided selection on existing parallel data and single-sided selection on monolingual data**, thereby avoiding large-scale translation, improving the efficiency of parallel data construction, and enhancing the data efficiency of cross-lingual transfer during training.
>
> **We sincerely hope that you will consider ParaRater’s contribution to the study of parallel data and cross-lingual transfer.**
>
> > **Q2**: Missing Baseline Comparison: The paper omits a critical baseline: random selection of English sentences for translation.
> >
>
> **A2:**
>
> We report the random selection results. The table below shows the averages across languages for each benchmark.
>
> | Benchmark | Random | ParaRater |
> | --- | --- | --- |
> | **ARC-E** | 44.39 | 45.97 |
> | **MMMLU** | 27.14 | 27.50 |
> | **BMLAMA** | 50.19 | 51.44 |
> | **HellaSwag** | 38.64 | 40.22 |
> | **StoryCloze** | 61.96 | 63.74 |
> | **XWinograd** | 51.45 | 62.30 |
> | **Flores** | 15.42 | 21.88 |
>
> **Random selection is a very weak baseline**, and this has been thoroughly demonstrated in many prior studies on data selection. Therefore, we chose two strong data selectors, DCLM and FineWeb-Edu, to compare with ParaRater. In our experiments, each selection method modifies only a relatively small portion of the data, 1B tokens per language, so the model performances remain relatively close. **If the proportion of parallel data were increased, the disadvantages of random selection would become much more pronounced.** Paper [1] also compares model performance when trained on FineWeb-Edu versus directly on the raw FineWeb2 dataset (Figures 10 and 11). As shown, using unfiltered data significantly degrades model performance.
>
> [1] Penedo, Guilherme, et al. "The fineweb datasets: Decanting the web for the finest text data at scale." *Advances in Neural Information Processing Systems* 37 (2024): 30811-30849.
>
> > **Q3:** Scalability and Transferability Concerns: It remains unclear whether the ParaRater selector must be retrained for different model architectures or scales (e.g., 7B vs. 70B LLMs).
> >
>
> **A3:**
>
> **Experiments at the pretraining stage require extremely large computational resources.** Validating on 1.2B models is already a relatively large setting compared to this type of study. Due to limited computational resources, we have not yet results on larger models but are actively pushing forward.
>
> **We have released all our raters, data, and meta-learning training code to facilitate reproducibility and to facilitate further research in this direction.**

---

> ### Author Response · Authors · 2025-11-21
>
> > **Q4:** Evaluation Scope Limited to Pretraining: The authors argue that parallel data is most effective during pretraining. However, given the prevalence of strong multilingual foundation models today, the community increasingly focuses on post-training (e.g., instruction tuning, alignment) for cross-lingual enhancement.
> >
>
> **A4:**
>
> **It is important to clarify that we did not claimed that *parallel data is most effective during pretraining*.** Our decision to conduct experiments at the pretraining stage is based on two considerations.
>
> 1. **Pretraining is the most critical phase in which a model develops its multilingual abilities**, yet prior work has paid very little attention to how parallel data affects this process and how cross-lingual transfer emerges during pretraining.
> 2. **Post-training investigations of cross-lingual transfer are influenced by the base model’s existing multilingual competence**, which introduces confounding factors.
>
> **Therefore, conducting experiments during pretraining enables a fully fair comparison and also fills a significant gap in understanding multilingual transfer at this stage.** This is one of the key contributions of our paper. Of course, applying ParaRater at the post-training stage is also an important direction for our future work. In future work, we plan first to investigate its effects during Continual Pretraining, and then explore more training paradigms such as SFT and RL.
>
> > **Q5:** Lack of Scaling Analysis: While the method is validated on a 1.2B-parameter model, its effectiveness on larger-scale LLMs (e.g., 7B, 13B, or beyond) is unexamined.
> >
>
> **A5:**
>
> This question is similar to Q3. We are actively working toward conducting larger-scale experiments.
>
> We note that many data studies [1,2] successfully use small models to estimate the effectiveness of data selection on performance trends of larger models—some even use models with only 1M parameters [1]. This reflects the strong structural commonality within Transformer models. **Therefore, demonstrating effectiveness at the 1.2B scale makes the results quite promising for extension to larger models such as 7B.**
>
> Following recent experimental practices in research on training data [3,4], our pretraining experiments at the 1.2B scale are already among the large settings. This scale is sufficient to draw convincing conclusions.
>
> [1] Liu, Fengze, et al. "Quadmix: Quality-diversity balanced data selection for efficient llm pretraining." *arXiv preprint arXiv:2504.16511* (2025).
>
> [2] Guo, Ping, et al. "Exploring Polyglot Harmony: On Multilingual Data Allocation for Large Language Models Pretraining." *arXiv preprint arXiv:2509.15556* (2025).
>
> [3] Muhammad Reza Qorib, Junyi Li, and Hwee Tou Ng. 2025. Just Go Parallel: Improving the Multilingual Capabilities of Large Language Models. In *Proceedings of the 63rd Annual Meeting of the Association for Computational Linguistics (Volume 1: Long Papers)*, pages 33411–33424, Vienna, Austria. Association for Computational Linguistics.
>
> [4] Calian, Dan A., et al. "DataRater: Meta-Learned Dataset Curation." *arXiv preprint arXiv:2505.17895* (2025).
>
>
> Once again, thank you for your valuable feedback. We sincerely hope that you will consider the novelty of our work and the contribution it makes to the multilingual research community.

---

> > ### Author Response · Authors · 2025-11-25
> >
> > In response to your concerns (Q3 and Q5) about **scalability and generalization**, we would like to provide additional experiments on a **3B model**. We use the same data configuration as before, with the only change being an increase in model size to 3B parameters. The following results are averaged across all languages.
> >
> > | **Benchmark** | **Fineweb-edu** | **ParaRater** |
> > | --- | --- | --- |
> > | **ARC-E** | **48.65** | 47.20 |
> > | **BMLAMA** | 51.97 | **52.69** |
> > | **HellaSwag** | 39.87 | **41.70** |
> > | **MMMLU** | 27.68 | 27.64 |
> > | **StoryCloze** | 63.45 | **64.30** |
> > | **XWinograd** | 63.55 | **64.13** |
> > | **Flores** | 19.84 | **21.12** |
> >
> > The observations on the 3B model are consistent with those on the 1.2B model, indicating that ParaRater’s data selection generalizes well. ParaRater delivers substantial improvements on NLU tasks and translation tasks. Since ParaRater is not trained with an objective that favors scientific data, its overrall performance on ARC-E is lower than FineWeb-Edu. However, it preserves English capabilities on ARC-E better (ParaRater **66.26** vs. Fineweb-Edu 64.9). Due to computational constraints, we are still actively working toward conducting experiments at larger models.
> >
> > Hope our answer solves your concern. Thank you again for your valuable suggestions.

---

### Note · Authors · 2025-12-05

I have read and agree with the venue's withdrawal policy on behalf of myself and my co-authors.